# A New Way of Rice Breeding: Polyploid Rice Breeding

**DOI:** 10.3390/plants10030422

**Published:** 2021-02-24

**Authors:** Rongrong Chen, Ziyi Feng, Xianhua Zhang, Zhaojian Song, Detian Cai

**Affiliations:** 1School of Life Sciences, Hubei University, Wuhan 430062, China; 201911110711104@stu.hubu.edu.cn (R.C.); 201911110711085@stu.hubu.edu.cn (Z.F.); 20050040@hubu.edu.cn (Z.S.); 20040640@hubu.edu.cn (D.C.); 2Wuhan Polyploid Bio-Technology Co., Ltd., Wuhan 430345, China

**Keywords:** rice, polyploid, polyploidization, polyploid rice

## Abstract

Polyploid rice, first discovered by Japanese scientist Eiiti Nakamori in 1933, has a history of nearly 90 years. In the following years, polyploid rice studies have mainly focused on innovations in breeding theory, induction technology and the creation of new germplasm, the analysis of agronomic traits and nutritional components, the study of gametophyte development and reproduction characteristics, DNA methylation modification and gene expression regulation, distant hybridization and utilization among subspecies, species and genomes. In recent years, *PMeS* lines and neo-tetraploid rice lines with stable high seed setting rate characteristics have been successively selected, breaking through the bottleneck of low seed setting rate of polyploid rice. Following, a series of theoretical and applied studies on high seed setting rate tetraploid rice were carried out. This has pushed research on polyploid rice to a new stage, opening new prospects for polyploid rice breeding.

## 1. Introduction

A polyploid individual is an individual with three or more sets of chromosomes in somatic cells. Polyploidization is one of the important ways of plant evolution. All flowering plants have experienced at least one polyploidization event during their evolutionary history [1]. Polyploid plants are characterized by large size, high nutrient and secondary metabolite content. They also show strong vitality and adaptability, drought and cold resistance, and other advantages [2]. Therefore, polyploid technology has been widely used in plant breeding, especially for the purpose of increasing the mass of vegetative organs or total biomass. In addition, polyploids can be used to breed varieties directly and can also serve as a bridge to overcome the barriers to distant hybridization and transfer foreign genes, thereby promoting gene exchange between species or populations [3]. At present, the most widely cultivated rice is diploid. Compared with polyploid crops (such as wheat), rice has a smaller genome and lower DNA content. The genetic resources of cultivated diploid rice are limited. Therefore, further development of rice breeding is hindered [4]. Compared with diploid rice, polyploid rice has advantageous agronomic traits which have attracted the widespread attention of rice researchers [5,6,7,8], such as large grain size, high 1000-grain weight, strong stem, strong stress tolerance, and high adaptability. Polyploid rice was first discovered and reported by Nakamori in 1933 [9]. A surge in polyploid rice breeding and research followed this discovery. Before the 1960s, scientists from Japan, India, and the Philippines had carried out research on autotetraploid rice, mainly focusing on genetics and applications, hybridization between cultivated and wild rice, and the evolutionary relationships of *Oryza* species. However, the bottleneck problem of low seed setting rate of autotetraploid rice had not been solved, which made it difficult to translate research results to production. As a result, most scientists stopped their research in this area [10]. In China, polyploid rice breeding was first studied by Bao et al. in 1951 [11]. After years of research, more than 150 autotetraploid rice germplasms were induced. Breeding goals and selection criteria for main agronomic traits in autotetraploid rice were put forward. The influence of polyploidization on rice agronomic traits and nutritional components was reviewed, and the anther culture and establishment of mutant populations of autotetraploid rice were studied [12,13,14,15,16]. These studies have accumulated valuable experience for polyploid rice research. In recent decades, researchers have furthered polyploid rice breeding and theoretical research. Huang et al. studied the gametophyte development and reproductive characteristics of autotetraploid rice, polyploid rice induction through low energy nitrogen ion beam, and distant hybridization between autotetraploid rice (*Oryza sativa*) and *Pennisetum alopecuroides*, *Leersia hexandra,* or *Oryza glaberrima* [17,18,19,20,21,22,23,24]. Tu et al. made useful explorations in the use of autotetraploid three-line hybrid rice, and studied the methylation modification and *Wx* gene expression of autotetraploid rice [25,26,27,28]. Liu et al. researched the reasons for the decrease of fertility and seed setting rate of autotetraploid rice through reproductive studies of autotetraploid rice and their interspecific hybrids [29,30,31,32]. Xing et al. used a basic population of double embryo seedlings to screen out a batch of autopolyploid rice lines and crossed them with normal diploids for early-generation stable breeding [33,34]. Yan et al. obtained double diploid plants resistant to brown planthopper from the anther culture progeny of the hybrid F_1_ of tight panicle wild rice and cultivated rice [35]. Chen et al. doubled naked rice, a new rice germplasm derived from distant hybridization between rice and wheat, into an autotetraploid [36]. Cai et al. proposed a breeding strategy of “selecting super rice by using the dual advantages of distant hybridization and polyploidy” based on insights from natural plant evolution [37]. They also established a high efficiency polyploid induction technology by combining tissue culture with chemical reagent treatment. Two highly fertile tetraploid rice lines with polyploid meiosis stability (*PMeS*) genes were bred, which broke through the bottleneck problem of low seed setting rate of polyploid rice for the first time. A batch of polyploid *indica-japonica* hybrids with heterosis were selected and successfully planted [38,39,40]. The PMeS lines and their hybrids have stable meiotic behaviors, normal embryonic development and high seed setting rates, characteristics which have laid a foundation for the genetic breeding and research of polyploid rice [41]. In recent years, another neo-tetraploid rice line with higher than 80% seed setting rate was selected through extensive hybridization between different types of autotetraploid rice and directional selection of high-fertility materials in hybrid progenies [42,43,44]. The neo-tetraploid rice also has high fertility. Moreover, it can overcome the sterility of most tetraploid rice hybrids and produce heterosis [45]. In addition, materials with special reproductive characteristics can be found in autotriploid rice, and apomixis materials are more likely to be obtained in polyploid rice than in diploid rice [46,47]. This previous research on polyploid rice has greatly enriched the knowledge in the field of plant polyploidy, and laid a foundation for the successful application of polyploid rice in production in the future.

## 2. Induction Methods of Polyploid Rice

From the relationships between plant evolution and their genomes, we know that polyploidization results in new species. Before human beings knew about polyploidy, most polyploids were produced by natural means. Because Blakeslee confirmed that colchicine can induce polyploidy in 1937, artificial polyploid induction has entered a new stage [48]. Research involves more and more plants, and research methods are constantly being improved and expanded [49,50]. Artificial induction of polyploidy is divided into physical induction, biological induction and chemical induction. In physical induction, temperature shocks, ionizing radiation, radiation, grafting, etc., can all lead to the formation of polyploids. Researchers have tried to use physical methods of rice polyploid induction. For example, Huang et al. used low energy N^+^ to inject autotetraploid rice, and selected a multi-embryonic seedling mutant in its M2 generation [17,51,52,53,54]. However, due to its low efficiency, the physical induction method is not widely used [55]. At present, chemical induction is the most widely used method of artificial polyploid induction. The commonly used chemical mutagens are colchicine, naphthalene pentane, naphthalene ethane, with colchicine being the most commonly used. Colchicine can inhibit the formation of spindles—the chromosomes separate but the cells do not divide. Thus, cells with increased ploidy are produced. Traditional chemical induction of polyploidy is to soak seedlings, buds or anthers with colchicine. In the early stage, this method was often used for polyploid induction in rice. For example, Bao and Yan treated diploid seedlings with colchicine to obtain autotetraploid rice [11]. Huang et al. used colchicine to treat buds to obtain polyploid rice [17,56,57,58]. A combination of low agar dosage, small inoculum dosage, large culture container, and more medium dosage is very beneficial to the rapid proliferation of adventitious buds of tetraploid rice in vitro [59]. However, the efficiency of doubling with seedlings, buds, or anthers is very low and chimeras are often formed. With the development of tissue culture technology, colchicine has been used to treat plant tissues or cells (such as callus, cluster buds, embryoids, protoplasts, etc.) in vitro. Other researchers have attempted to induce polyploidy by plant regeneration [49]. Compared with traditional methods, plant regeneration methods lead to more efficiency and less chimerism [60,61]. Cai et al. used tissue culture and colchicine induction to obtain polyploid rice with high efficiency [61,62,63]. Their method was to use rice seed embryos or young panicles to induce vigorous growth callus. The callus was then treated with colchicine for a specific time and at a specific concentration. This resulted in the induction of seedlings. Finally, polyploid rice was confirmed by ploidy test. Jiang et al. used five diploid rice varieties as materials for a doubling experiment. In the process of in vitro culture of young panicle, the callus was treated with colchicine. The resulting average induction rate of tetraploidy was 46% [64]. Four diploid *indica-japonica* hybrids were studied by Huang et al. [65] They treated calli and germinating seeds with colchicine. The highest doubling rate of callus treated with colchicine was 56.3%. The average doubling rate was about 10 times higher than that of germinated seeds. Diploid hybrids of an extreme type of *indica-japonica* subspecies were induced by Liu et al. [66]. The calli, seedlings, and germinating seeds were each treated with colchicine. The authors found that the treatment of callus had the highest doubling rate and efficiency. Wang et al. found that the induction efficiency of autotetraploid rice is related to many factors, such as the genetic type of induction material, concentration of colchicine solution, induction time, etc. Among these factors, genetic type was the most important [55]. The induction efficiencies of different types of materials under the same conditions usually showed differences. Therefore, the doubling conditions of different types of materials need to be finely adjusted. Luan et al. also found that the doubling rate of different materials was significantly different when they used colchicine to treat rice callus to create autotetraploid rice [67]. Xiao et al. found that 70% thiophanate-methyl has a significant promoting effect on the growth and rapid proliferation of tetraploid rice test-tube seedlings, which can greatly improve reproduction efficiency and reduce costs [68]. In recent years, some new inhibitors have been used in plant-induced doubling [60]. In addition, Sun et al. also obtained polyploid rice through somaclonal variation. The percentages of multiploids occurring in somaclones ranged from 0–13.3% in nine varieties (or hybrids) of *indica* group, but no multiploid was found in nine varieties (or hybrid) of *japonica* group [69].

## 3. Autotetraploid Rice

Autopolyploidy, in which the added genomes come from the same species, is produced by direct doubling of diploid chromosomes. It is common in plants [70]. Although polyploidization often also involves hybridization (allopolyploidy), it is clear that doubling the genome, in and of itself (autopolyploidy), is a macromutation: a single event with many phenotypic consequences [71]. Artificial autotetraploid rice can be obtained by artificially treating rice seeds, seedlings or calli with physical or chemical methods [11,17,72]. The grain size of autotetraploid rice is larger and heavier than that of diploid rice. At the same time, grain bulk density is stable and has great potential to increase yield [73]. However, compared with diploid rice, autotetraploid rice has weaker sexual reproduction ability and lower seed setting rate. This problem has been perplexing breeders, resulting in the slow progress of polyploid rice breeding research [74,75,76,77,78,79]. Bao et al. made long-term efforts to improve autopolyploid rice, obtaining excellent autopolyploid strains. These autotetraploid rice lines have the characteristics of relatively high seed setting rates, short plant types, and high yields [80,81]. These researchers also carried out beneficial explorations in induction methods, yield potential, stability of individual plant characters, protein content of grains, screening of primary trisomics, and hybridization between cultivated rice and wild rice with different ploidy [18,82]. There are more and more studies on heterologous addition lines obtained by crossing autotetraploid rice with wild rice [83]. Hybridization between autotetraploid and wild species can improve the fertility of F_1_ and allow to easily obtain heterologous addition lines. In recent years, studies on autotetraploid rice have mainly focused on photosynthetic characteristics, quality characteristics, DNA methylation, agronomic traits, seed setting rate, and yield potential. Research has also focused on the growth and development characteristics of autotetraploid rice and the law of heredity of its offspring. Moreover, researchers have been trying to overcome the main shortcomings of low seed setting rate, and directly use the advantages of polyploidy in grain quality, resistance, and biological yield, so as to study and utilize autotetraploid rice [19,84,85,86]. In their study of photosynthetic characteristics, Ruan et al. found that the photosynthetic rate of autotetraploid rice was higher than that of diploid rice at the late growth stage [87]. Under drought conditions, the photosynthetic rate of autotetraploid rice gave it a certain advantage. The chlorophyll content of the flag leaf, the maximum quantum yield (Fv/Fm) and the actual quantum yield of PSⅡ(ΦPSⅡ) remained relatively high. Xie et al. measured the light sum curve of flag leaves of diploid and autotetraploid rice under continuous natural light at grain filling stage [88,89]. The light compensation point of diploid rice was about 2.5 klux, and that of tetraploid rice was about 4 klux. Their net photosynthetic rates fluctuated with the light intensity, but their fluctuation amplitude was different from each other. The net photosynthetic rate of flag leaves of autotetraploid rice at heading stage was significantly lower than that of diploid rice under strong illumination (higher than 40 klux). Under weak illumination (15–20 klux), the net photosynthetic rate of tetraploid was similar to that of diploid rice, but the difference was not significant. The net photosynthetic rate of tetraploid was higher than that of diploid rice under extremely weak illumination (below 15 klux). This indicates that there are differences in photosynthetic characteristics between autotetraploid and diploid rice. The change of ploidy has an obvious effect on photosynthetic characteristics. Yang et al. also found that autotetraploid rice lines had a high photosynthetic capacity and high light-utilization efficiency [90]. In fact, the relationship between ploidy and photosynthetic parameters is very complex. With the increase of ploidy, some characteristics improve significantly, but some characters do not change significantly [91]. In the study of nutritional quality, Song et al. found that the protein content and total amino acid content of tetraploid rice were generally higher. Amylose content was also low [14]. Compared with the corresponding diploid rice varieties, the tetraploid rice varieties are more nutritious and palatable. Xie et al. analyzed the specific expression results of autotetraploid rice at protein level [92]. They showed that the total protein content of autotetraploid rice GUI 630 was significantly higher than that of diploid rice GUI 630. In addition, researchers found that genome doubling did not significantly change the genetic relationship between *Wx* gene and rice. However, the correlation coefficient between autotetraploid rice and *Wx* genotype was lower than that of diploid rice. It is speculated that genome doubling may affect the original gene expression regulation. Microsatellite marker and molecular marker analysis showed that the *Wx* gene locus changed in some autotetraploid rice with amylose content changed. At the same time, the amylose content of autotetraploid rice was negatively correlated with plant height and grain width, and positively correlated with other agronomic traits. Among them, there was significant correlation with flag leaf length, 1000-grain weight, and theoretical yield, but extremely significant correlation with grain length [26,93,94,95]. Song et al. also pointed out that waxy endosperm was inherited by single gene recessive inheritance [96,97]. There are four alleles at *Wx* locus in autotetraploid glutinous rice, so the range of genotypes and phenotypes of hybrids between autotetraploid glutinous rice and non-glutinous rice is much more complex than that of diploid rice. Yang et al. researched the mechanism of starch content increase in grain of autotetraploid rice [90]. They found that the activities of ADP-glucose pyrophosphorylase, soluble starch synthase, and starch-branching enzyme in grains of autotetraploid rice lines were higher than those in grains of corresponding diploid rice lines during the grain-filling stage. Therefore, autotetraploid rice lines were more efficient than corresponding diploid rice lines in converting photosynthetic products into starch. In the study of DNA methylation of autotetraploid rice, Yang et al. found that the average total methylation rate of autotetraploid rice was significantly lower than that of diploid materials [28]. The two materials also had great differences in the proportion of half- and fully methylated DNA. However, the degree of methylation did not increase due to polyploidization. After further analysis of the characteristics of DNA methylation patterns of rice of different ploidy, it was found that the proportion of hypermethylated genes in autotetraploid rice was significantly higher than the proportion of methylated genes. It was suggested that the expression of some functional genes in autotetraploid rice may be inhibited by methylation, which alleviates the dose effect of genome doubling. In the study of agronomic traits, Nakamori [9] was the first to describe the characteristics of autotetraploid rice, such as large grain and awn. The agronomic characteristics of autotetraploid rice and diploid rice were compared in detail by Song et al. [14]. The obvious characteristics of tetraploid rice were a decrease in panicle number, grain number per panicle, seed setting rate and plant height, but an increase in 1000-grain weight, thicker stem, and higher lodging resistance. Cui et al. found that the expression of seed size-related genes in autotetraploid rice were related to ploidy, developmental stages, and gene functions [98]. Seed size could be increased by controlling the expression of seed size genes.

## 4. *Indica-Japonica* Tetraploid Hybrid Rice

Asian cultivated rice is divided into two subspecies, *indica* and *japonica*. There are obvious differences in morphological, physiological, and biochemical characteristics between them. Studies have shown that the heterosis between *indica* and *japonica* is much greater than that within *indica* or *japonica*. The utilization of heterosis between *indica* and *japonica* is an important new approach for rice breeding. From the perspective of genomics, there are significant differences in genome sizes, gene numbers and gene categories between *indica* and *japonica* [99,100,101,102,103]. Under diploid condition, this not only affects the normal chromosome pairing during meiosis, but also the gene expression at each developmental stage, which makes *indica-japonica* hybrids difficult to use. However, under tetraploid conditions, this imbalance can be overcome, which is conducive to the utilization of the heterosis of *indica* and *japonica* rice. Huang et al. compared diploid rice, autotetraploid rice and *indica-japonica* tetraploid hybrids [104]. The plant height, growth period, panicle length, effective panicle per plant, spikelets per panicle and 1000-grain weight of *indica-japonica* tetraploid hybrid rice were significantly higher. The number of primary branches and secondary branches were also higher. They found no significant differences in the number of spikelets per plant and spikelets per panicle among different ploidy *indica-japonica* hybrids. However, there were significant differences in 1000-grain weight and seed setting rates. In addition, during the development of male gametes and gametophytes, the F_2_ population of tetraploid *indica-japonica* hybrid were normal [105,106]. Using diploid and tetraploid *indica-japonica* hybrids as materials, Chen et al. also found that the effects on seed setting rates varied with different hybrids. Generally, the seed setting rate of tetraploid *indica-japonica* hybrid was significantly higher than that of diploid rice [107]. Cai et al. put forward a strategy of tetraploid *indica-japonica* hybrid breeding by selecting tetraploid *indica* and *japonica* parents with distant genetic relationships, using Ph-like genes which can inhibit partial homologous chromosome pairing [37,108]. The authors went on to breed highly fertile tetraploid rice lines with polyploid meiosis stability (*PMeS*) genes. With *PMeS* lines as parents, a number of strong heterosis *indica-japonica* tetraploid hybrids were obtained. For example, tetraploid hybrids PSR073 and PSR120 showed obvious advantages in a variety of agronomic traits. Compared with diploid rice, they also had stronger resistance to adverse environmental conditions [38,109].

The hybrids of typical *indica* and typical *japonica* usually show obvious biological yield heterosis effect. Using tetraploid *indica-japonica* hybrid as a bridge is a way to tap into the greater yield potential of rice [110]. Fan et al. showed a new way to study the law of heredity of *indica-japonica* hybrid and to carry out rice breeding by doubling the diploid *indica-japonica* hybrids into tetraploid, and then inducing double haploidy through another culture [111]. The doubled *indica-japonica* hybrids had more favorable genetic background than those without doubling. Amphihaploid rice could overcome many obstacles of *indica-japonica* hybrids and has potential value in breeding. Wu et al. studied the cause of the large phenotypic diversity in the F_4_ population of newly synthesized *japonica-indica* tetraploid rice [112]. Aneuploidy, as an unavoidable intermediate in the process of new polyploid formation, can transmit the characteristics of the present generation to the offspring of aneuploid rice. This could promote the diversity of the populations and their adaptability to new environments. In *japonica-indica* tetraploid rice induced by colchicine, chimera of aneuploid and euploid rice appeared in some somatic cells [63].

## 5. Polyploid Male Sterile Rice Lines

In the utilization of plant heterosis, sterile lines are the key germplasms. In rice production, the use of heterosis to improve rice yield largely depends on the breeding of male sterile lines. In 1964, Yuan et al.’s search for natural male sterile mutants of rice marked the beginning of the research on male sterile lines in rice [113]. In the study of polyploid rice male sterile lines, Tan et al. [114] proposed in 1976 the idea of breeding polyploid rice three-line sterile lines and formulating polyploid rice hybrids. Tu et al. bred autotetraploid male sterile lines, maintainer lines and restorer lines, thus realizing the three-line matching of autotetraploid rice [25]. Additionally, some new autotetraploid hybrid rice combinations showing great yield potential have also been preliminarily bred. In their study of two-line male sterile line of polyploid rice, Zhang et al. bred two photo-thermo-sensitive genic male sterile lines of polyploid rice for the first time, PS006 and PS012, through chromosome doubling, compound hybridization, and selfing [115]. The two male sterile lines had good flowering habits, high outcrossing rates, obvious changes in fertility, and good combining abilities. Their hybrids showed strong heterosis, with the potential to increase yields and improve crop quality. They provided germplasm resources for studying and utilizing the heterosis of polyploid rice. In their research on the agronomic characteristics and pollen fertility of polyploid rice sterile lines, Ma et al. compared diploid and tetraploid photo-thermo-sensitive genic male sterile Peiai 64 S rice lines. The glumes, anthers, stigmas, and angles of the flag leaf of tetraploid rice were significantly larger than those of diploid rice, but the length and width of the flag leaf of tetraploid rice was significantly lower [116]. The pollen fertility of tetraploid Peiai 64 S was similar to that of the diploid line. It was sterile under long-day and high-temperature conditions, but fertile under short-day and low-temperature conditions. Liu et al. studied the growth characteristics, flowering habits, and pollen organs of male sterile and restorer lines of polyploid rice [117,118]. They found that chromosome doubling had a certain effect on many characteristics, but had no significant effect on sterility and restorability. The tetraploid restorer line was characterized by large pollen grains and a large amount of pollen. Tetraploid male sterile lines were characterized by large pollen grains, small variation of pollen amount, and stable pollen sterility. These characteristics are helpful to utilize the heterosis of polyploid rice.

## 6. Allopolyploid Rice

Allopolyploidy is a kind of polyploidy formed by chromosome doubling of hybrids between different species. Allopolyploidization leads to heterozygosity between genomes, which can increase the genome capacities, widen the ranges of genetic variation, and enhance tolerance to adverse environmental factors. It has obvious advantages in genetic evolution. After allopolyploidization, fertility can often be restored completely or partially, which plays an important role in overcoming the sterility of distant hybridization [71]. Therefore, allopolyploidy as a genetic medium has become an effective means for gene transfer or introgression [4]. Most of the natural polyploid species that appear between related species are allopolyploid [80]. There are many varieties of heteropolyploid rice in nature, such as *Oryza minuta* (BBCC), *Oryza latifolia* (CCDD), *Oryza coarctata* (HHLL)*, Oryza longiglumis Jansen* (HHJJ), *Oryza schlechteri Pilger* (HHKK), etc. [119]. Allopolyploid rices are important resources for biological research and rice breeding. The combination of distant hybridization and polyploidization is promising for rice breeding [120]. Artificial allopolyploids can be obtained by crossing cultivated rice with wild rice and doubling their chromosomes. Zhang et al. established an efficient technology, the development of synthetic allopolyploid rice, to obtain allopolyploid rice based on a wild cross, embryo rescue, and in vitro colchicine treatment [121]. Using this technology, fourteen typological allopolyploid rice lines have been developed from six interspecies and seven different genomic hybridizations. Compared with amphihaploid rice (AG), the plant height, flag leaf length, number of spikelets, and glume length/width of artificial allotetraploid rice (AAGG) significantly increased, giving obvious advantages to the AAGG line. Moreover, the resulting allotetraploid rice (AAGG) was awnless, a characteristic which has high utilization value [122]. Synthetic allotetraploid rice (AABB) showed obvious superiority in growth and production. It had normal meiosis and fertility, and was an excellent germplasm for genetic evolution research and rice breeding research [4]. Zhu et al. obtained the hybrid triploid ACD of cultivated rice (AA) and *Oryza alta* (CCDD) through distant hybridization [123]. Then, the allohexaploid rice AACCDD was obtained by chromosome doubling. Studies have shown that there are significant differences in reproductive mode and outcome between allotriploids and allohexaploids, as well as differences between allohexaploid females and males. In order to overcome the sterility of the allohexaploid rice AACCDD and verify the female fertility of the hybrid, an allooctoploid AAAACCDD was created to obtain a sturdy allopolyploid hybrid.

In their study of the genetic characteristics of allopolyploid rice, Wu et al. resequenced the whole genomes of heteropolyploid rice [124]. In the early segmental allotetraploid population, 40% of the inbred plants were composed of 55 different aneuploidy karyotypes. It was more common to gain a chromosome than to lose a chromosome. The 12 rice chromosomes had obvious aneuploidy. In addition, at the population level, segmental allotetraploidy produced widespread homologous recombination and phenotypic diversity, which may be an important feature of the evolution of allotetraploid [125]. Wang et al. found that there was no selective bias in the synergism between nucleus and cytoplasm in young allopolyploid rice [126]. In addition to DNA sequence and gene expression, more complex regulatory mechanisms may be necessary to achieve the ultimate nucleocytoplasmic synergy.

## 7. The Molecular Mechanism of Polyploid Rice Fertility

Polyploid rice shows strong biological advantages and yield potential, but its low seed setting rate is the main obstacle to its wide agronomic utilization. Researchers found that the abnormal meiosis behavior of polyploid rice is the main reason for its poor pollen fertility, low seed setting rate, and even sterility [38]. During the reproductive development of polyploid rice, microRNA often plays an important regulatory role. Using high-throughput sequencing technology, Li et al. analyzed microRNA during the pollen development of diploid and autotetraploid rice. In autotetraploid rice, 57 microRNAs were specifically expressed [127]. A large number of 24 nt siRNAs related to transposable factors were found during the pollen development phase of autotetraploid rice, but the number of transposable factors was significantly lower in diploid rice. This indicates that 24 nt siRNA may play a role in pollen development. The cytological variation and the changes of small RNA during the development of pollen and embryo sac in autotetraploid rice were further analyzed by Li et al. [128]. A large number of differentially expressed microRNAs were detected, along with 21 nt-phasi RNAs. The 24 nt-phasi RNAs were only found during pollen development. Compared with diploid rice, the expression of 24 nt-phasiRNAs was down-regulated in autotetraploid rice, which could lead to abnormal pollen mother cells of autotetraploid rice. The authors also found that the expression of 24 nt TEs-siRNAs was up-regulated in the embryo sac, but down-regulated during pollen development. In autotetraploid rice, 24 nt TEs-siRNAs could cause genome instability and sterility. On the other hand, Li et al. pointed out that the differential expression level of lncRNAs was related to transposable factors and meiotic regulatory targets, which may be the endogenous noncoding regulatory factors of pollen/embryo sac development in autotetraploid rice [129,130]. They also found that the low fertility of autotetraploid rice was caused not only by the differential expression of pollen development related genes, but also by sequence variation and differential methylation. Moreover, Lu et al. proved that *MOF1a* deficiency changes the gene expressions of pollen development and meiotic chromosome behavior of tetraploid rice, resulting in low pollen fertility of tetraploid rice [131]. In autotetraploid hybrids, Wu et al. found that the interaction of Sa, Sb, and Sc pollen sterile sites led to severe pollen sterility in autotetraploid rice, which resulted in the down-regulation of important meiosis related genes and transcription regulators [5,132]. Other studies on the fertility of tetraploid rice showed that polyploidization not only induced meiosis related genes expression mutations, resulting in abnormal chromosome behavior, but also caused changes in carbohydrate distribution and carbohydrate-related gene expression patterns. All of these factors led to the sterility of autotetraploid rice [133]. Abnormal microtubule distribution was also an important factor affecting pollen fertility and seed setting rate of autotetraploid rice, and was closely related to chromosome behaviors [134]. All the above studies indicate that the mechanism of pollen sterility in autotetraploid rice is complex and may be affected by many factors.

In recent years, tetraploid *PMeS* rice lines and neo-tetraploid rice lines with stable high seed setting characteristics have been successfully selected. They have provided valuable materials for the studies of reproductive development and fertility mechanisms of polyploid rice, and also for the breeding application of polyploid rice [30,38]. Xiong et al. studied the fertility mechanism of *PMeS* lines [135]. They found that *OsMND1* gene plays an important role in stabilizing polyploid rice meiosis behaviors, improving pollen fertility and influencing the expression of meiosis related genes. *OsMND1* participates in stabilizing meiosis by maintaining the balance of pairing, synapsis, and recombination during early meiosis. Overexpression of *OsMND1* can improve pollen fertility and viability, early normal embryonic development and seed setting rate of autotetraploid rice. However, RNAi of *OsMND1* in *PMeS* strains hinder the normal development of pollen and embryo, and reduce fertility. In research on fertility characteristics of new tetraploid rice, Bei et al. found that genome structure recombinations, epigenetic recombinations, DNA variations, meiosis and mutations of epigenetic related genes were all related to the high fertility of neo-tetraploid rice [32]. Chen et al. studied the heterosis and potential molecular mechanism of the hybridization of autotetraploid and neo-tetraploid rice [31]. The high expression of carbohydrate metabolism and fertility related genes could promote the heterosis of autotetraploid rice. Recently, Ghaleb et al. developed another long panicle neo-tetraploid rice called Huaduo 8, with natural fertility of pollen and embryo sac. This line also showed high fertility and heterosis when crossed with other low fertility autotetraploid rice [30,31,32,136].

## 8. Conclusions

Rice (*Oryza sativa* L.) is one of the world’s three major food crops. Faced with the pressure of a growing world population and the decrease in available arable land, it has become important to increase rice yield [4,115]. Now, this challenge becomes even greater owing to rapid climate changes, which has already become another immense threat to agriculture and food security. Therefore, there is an urgent need to develop new high yield rice varieties that are more resilient to climate variability and disasters [137]. Distant hybridization and polyploidization are important trends in plant evolution, as well as important directions in crop evolution. Polyploid plants often have advantages in genome buffering, vigorousness, and robust adaptation to environmental changes [138,139]. The genome of rice is small and its DNA content is low. However, the genetic resources of *Oryza* genus are very rich. Therefore, increasing the ploidy levels, increasing the genome capacities, and expanding the range of genetic variations are beneficial explorations for the basic theory and practical application of rice breeding. Since the first report of tetraploid rice in 1933, great achievements have been made in research on polyploid rice. In particular, the breeding of high-fertility PMeS tetraploid rice lines and neo-tetraploid rice lines broke through the bottleneck of low seed setting rate of polyploid rice, which made polyploid rice research enter a new stage of development. With the deepening of our understanding, we believe that polyploid rice breeding as a new way of breeding will show attractive prospects.

## Data Availability

Data sharing not applicable.

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
