# Peer review of "A New Way of Rice Breeding: Polyploid Rice Breeding"

_plants, 2021, doi:10.3390/plants10030422_

Round 1

Reviewer 1 Report

I think this is a well written and well referenced paper, and the references to previous work are helpful and set out in an a logical way. The conclusions are clear. Perhaps more could have been said about the general importance of this work, and how rice breeding is beneficial in terms of global food security, adapting agriculture to climate change, and the sustainable development goals to give some greater context to the study. See edits and suggestions in the attached document. 

There is a little incidence of non-native English that it would be helpful to correct, but nothing that is not understandable. 

Reviewer 2 Report

Review of “A New Way of Rice Breeding: Polyploid Rice Breeding” by Chen et al.

This review article details the main advances of polyploid breeding in rice with a particular emphasis on the use of PMeS lines as a means of making further progress in the future. The article’s discussion of the underlaying science seems sound and I believe would be useful to anyone wishing to gain a background in breeding polyploid rice. The article would benefit from the use of a figure or two, or perhaps a photo of heterosis, but this is not entirely necessary as there is an extensive bibliography.  The article’s English could use a little work but it was acceptable. I have made some suggestions for further improvement below.

Line 21: I suggest that the authors correct their statement that “Almost 100% of higher plants have experienced one or several polyploidization events”.  The fact is that we now know that ALL (100%) of flowering plants have experienced at least one polyploidization event during their evolutionary history (see Jiao et al., Nature. 2011 May 5;473(7345):97-100.

doi: 10.1038/nature09916).

Line 54: pennisetum needs to be capitalized and I think it should be italicized as well, i.e. Pennisetum, because this genus is not commonly known like Arabidopsis.

Lines 74, 78, etc…: The authors should try to avoid the use of words like “good” and “bad” as these tend to denote a bias. Better to use scientific terms like high, low, etc.

Line 79: as in above, is heterosis really obvious? Certainly, it is expected or even predicted but is it always obvious? Shouldn’t it be measured first? For an example of why heterosis is not always obvious, see Line 183. Maybe just delete that word here.

Line 142: I don’t understand how polysomic inheritance can be evolutionarily advantageous? Doesn’t that cause high rates of infertility until, and only until, the genomes become diploidized? Please explain or modify.

Line 278: two was misspelled as tow.

Line 318: the term allodiploid is new to me. I’m more familiar with amphihaploid or even polyhaploid for describe the haploid from a polyploid.

Line 326: as above, the term heterohexaploid is new to me. I suspect the authors mean allohexaploid AACCDD.

Lines 351 and 354: change SiRNAs to siRNAs.

Reviewer 3 Report

The article is very important for rice breeding. It is well written and presented. The bibliography is strong providing a State of the Art that is certainly useful for coming research projects and scientific publications. The sections well-developed and sustain the Conclusions

I found two papers that could enrich the article, but into my opinion the inclusion is not impeditive of the acceptance for publication  

Photosynthetica 2019, 57(2):680-687 | DOI: 10.32615/ps.2019.044

The mechanism of starch content increase in grain of autotetraploid rice (Oryza sativa L.)

P.M. YANG1, X.R. ZHOU1, Q.C. HUANG2

Theoretical and Applied Genetics, vol 67, pages67–73(1983)

Somaclonal genetics of rice, Oryza sativa L.. Theoret. Appl. Genetics 67, 67–73 (1983). https://doi.org/10.1007/BF00303925

Zong-xiu, S., Cheng-zhang, Z., Kang-le, Z. et al., November 1983
